# An Overview of the Role of Multiparametric MRI in the Investigation of Testicular Tumors

**DOI:** 10.3390/cancers14163912

**Published:** 2022-08-13

**Authors:** Athina C. Tsili, Nikolaos Sofikitis, Ourania Pappa, Christina K. Bougia, Maria I. Argyropoulou

**Affiliations:** 1Department of Clinical Radiology, Faculty of Medicine, School of Health Sciences, University of Ioannina, University Campus, 451 10 Ioannina, Greece; 2Department of Urology, Faculty of Medicine, School of Health Sciences, University of Ioannina, University Campus, 451 10 Ioannina, Greece

**Keywords:** magnetic resonance imaging, multiparametric magnetic resonance imaging, testis, testicular neoplasms

## Abstract

**Simple Summary:**

Although conventional ultrasonography remains the primary imaging modality for the assessment of testicular tumors, multiparametric MRI of the scrotum, which combines morphologic and functional data, serves as a powerful adjunct. Based on the recommendations issued by the Scrotal and Penile Imaging Working Group of the European Society of Urogenital Radiology, scrotal MRI is strongly recommended after equivocal US findings. In cases of testicular masses, the main clinical indications are as follows: lesion characterization when sonographic findings are non-diagnostic, discrimination between germ-cell and non-germ-cell testicular tumors, local staging of testicular tumors in patients who are candidates for testis-sparing surgery, and preoperative histological characterization of testicular germ-cell tumors in selected cases. This article aims to provide an overview of the role of multiparametric MRI in the investigation of testicular tumors.

**Abstract:**

Conventional ultrasonography represents the mainstay of testis imaging. In cases in which ultrasonography is inconclusive, scrotal MRI using a multiparametric protocol may be used as a useful problem-solving tool. MRI of the scrotum is primarily recommended for differentiating between benign and malignant testicular masses when sonographic findings are ambiguous. This technique is also accurate in the preoperative local staging of testicular tumors and, therefore, is recommended in patients scheduled for testis-sparing surgery. In addition, MRI may provide valuable information regarding the histological characterization of testicular germ-cell tumors, in selected cases. Scrotal MRI may also help in the differentiation between testicular germ-cell neoplasms and non-germ-cell neoplasms. Axial T1-weighted imaging, axial and coronal T2-weighted imaging, axial diffusion-weighted imaging, and coronal subtracted dynamic contrast-enhanced imaging are the minimum requirements for scrotal MRI. A variety of MRI techniques—including diffusion tensor imaging, magnetization transfer imaging, proton MR spectroscopy, volumetric apparent diffusion coefficient histogram analysis, and MRI-based radiomics—are being investigated for testicular mass characterization, providing valuable supplementary diagnostic information. In the present review, we aim to discuss clinical indications for scrotal MRI in cases of testicular tumors, along with MRI findings of common testicular malignancies.

## 1. Introduction

Scrotal ultrasonography (US), with a combination of grayscale and color Doppler US, is the first-line imaging technique for the evaluation of testicular pathology, due to its low cost, wide availability, convenience, short examination time, lack of radiation exposure, and high diagnostic accuracy [1,2,3,4,5,6,7]. This technique represents the initial imaging diagnostic tool for the confirmation of the presence of a testicular mass and for the evaluation of the contralateral testis, and is strongly recommended, even in the presence of a clinically evident testicular tumor [8]. However, US has limitations related to the operator’s practice and clinical experience, along with its relatively small field of view and lack of adequate tissue characterization. Occasionally, scrotal US findings are equivocal or nonspecific, including indeterminate nature of a testicular mass, difficulties in delineating lesions’ location and/or extension, and discrepancies between US findings and clinical history [1,2,3,4,5,6,7,9]. Multiparametric US—including conventional US, contrast-enhanced US, and elastography—has significantly improved the diagnostic efficacy of the technique in the assessment of scrotal diseases [10,11,12,13].

Multiparametric MRI, which combines morphological and functional information, has emerged as an accurate and cost-effective second-line imaging technique for the assessment of the scrotum, and is especially recommended in cases of inconclusive US findings [7,9,14,15,16,17,18,19,20,21,22,23,24,25,26,27,28,29,30,31,32,33,34,35,36,37,38]. MRI uses non-ionizing radiation, and is less operator-dependent than US. Scrotal MRI has a wide field of view and multiplanar capabilities, allowing the simultaneous assessment of both testes, along with paratesticular spaces and the spermatic cords and, therefore, a detailed depiction of the complex scrotal anatomy. MRI’s superior soft-tissue resolution characteristics combined with the ability of the technique to detect subtle enhancement provide valuable information in the characterization of scrotal masses. MRI findings, interpreted in conjunction with US features with respect to lesion location, morphology, tissue characteristics, and functional data, may help narrow differential diagnosis.

Scrotal MRI as an adjunct tool to US reliably helps in the detection, localization, and characterization of lesions, allowing for more precise treatment planning and for surgical exploration or radical orchiectomy to be avoided in some patients. This technique performs well in the differentiation between intratesticular and paratesticular masses, in the characterization of the nature of scrotal masses, and in the evaluation of the local extent of testicular carcinomas [7,9,14,15,16,17,18,19,20,21,22,23,24,25,26,27,28,29,30,31,32,33,34,35,36,37,38]. The Scrotal and Penile Imaging Working Group (SPIWG) appointed by the board of the European Society of Urogenital Radiology has produced recommendations on the optimal imaging techniques and clinical indications for scrotal MRI, based on literature review and the combined expertise of the group [14].

Testicular germ-cell tumors (TGCTs) comprise more than 95% of testicular malignancies. Histologically, these neoplasms are categorized into postpubertal tumors—including those derived from a “germ cell neoplasia in situ” (GCNIS)—and prepubertal tumors, including those unrelated to GCNIS [39]. The first group includes pure seminomas and nonseminomatous germ-cell tumors (NSGCTs), such as embryonal carcinomas, teratomas, and yolk sac tumors of postpubertal type; teratomas with somatic-type malignancies; and trophoblastic tumors, including choriocarcinoma, mixed NSGCTs, and regressed germ-cell tumors. TGCTs not derived from GCNIS include spermatocytic tumors, prepubertal-type teratomas (e.g., dermoid and epidermoid cysts, and well-differentiated neuroendocrine tumors), prepubertal mixed teratoma and yolk sac tumors, and prepubertal yolk sac tumors [39]. Sometimes, patients with TGCTs present with extensive metastatic disease, while the primary tumor has regressed (“burned-out” tumor (BOT)) [9].

Sex-cord stromal tumors represent the second most common group of testicular carcinomas, accounting for 4% of testicular tumors in the adult population. Leydig-cell tumors (LCTs) are the most common pure neoplasm in this category, followed by Sertoli-cell tumors (SCTs), granulosa-cell tumors, and pure stromal tumors. LCTs are benign tumors, although a small minority are clinically malignant [40]. Diffuse large B-cell lymphoma represents the most common testicular hematolymphoid malignancy [40]. Although rare, it is the most common testicular tumor in men 60 years of age and older [9,41].

Regarding testicular tumors, scrotal MRI is primarily recommended for the differentiation between benign and malignant testicular masses following indeterminate US findings [14]. This technique provides valuable information in the preoperative planning, local staging, and histologic characterization of TGCTs, in selected cases [14]. It may also help in the differentiation between germ-cell and non-germ-cell neoplasms, and specifically in the characterization of LCTs, allowing for the adoption of conservative surgery or active surveillance in compliant patients [14].

Routine multiparametric MRI (mpMRI) of the scrotum should include axial T1-weighted imaging (T1WI), axial and coronal T2-weighted imaging (T2WI), axial diffusion-weighted imaging (DWI), and coronal subtracted dynamic contrast-enhanced (DCE) imaging [14]. A variety of MRI techniques—including diffusion tensor imaging (DTI), magnetization transfer imaging (MTI), proton MR spectroscopy (1H-MRS), volumetric apparent diffusion coefficient (ADC) histogram analysis, and MRI-based radiomics—have recently been introduced, providing useful adjuvant diagnostic information in the characterization of testicular masses [42,43,44,45,46,47,48,49]. This article aims to provide an overview of the role of multiparametric MRI in the investigation of testicular tumors.

## 2. mpMRI Protocol of the Scrotum

Correct positioning is essential for scrotal MRI [14]. Patients are placed into the MRI unit in the supine position, with the feet first. The penis is raised and fixed to the lower abdominal wall, covered with a gauge. A towel is placed between the thighs to elevate the scrotum and to ensure that both testes have a similar distance from the coil. A circular surface coil is placed over the scrotum, on top of a second towel [14,22,29].

Thin-section, small-field-of-view T2WI in at least two orthogonal planes should be obtained, along the testes’ transverse axis and length. T2WI is accurate for the detection, localization, and characterization of lesions. The coronal plane allows a comparative evaluation of both testes, along with the paratesticular spaces and the spermatic cords. Sagittal T2WI may be added, and is helpful in lesion localization, assessment of small mass lesions, and local staging of testicular tumors [14]. Axial T1WI should be included in the routine MRI protocol, and provides information on scrotal anatomy and the presence of areas of increased signal, corresponding to either fat or methemoglobin [14]. DWI should be acquired in the axial plane, with at least three *b*-values (0, 400–500, and 800–1000 s/mm^2^). Apparent diffusion coefficient (ADC) maps should be created on the MRI console and analyzed both qualitatively and quantitatively. ADC calculation is performed using the highest *b*-value (800–1000 s/mm^2^) [14]. Coronal DCE imaging is performed after the bolus intravenous injection of gadolinium-based contrast medium. Five to seven consecutive imaging sets, each of 60 s duration, are acquired after the injection of the contrast material. The total examination time is 8 min. Subtraction is subsequently performed using the dataset obtained before administration of the contrast agent as a mask, to subtract slice-by-slice from each of the datasets obtained after contrast injection [14].

High magnetic fields (3.0 T) provide a higher signal-to-noise ratio compared to 1.5 T magnets, resulting in improvement of the diagnostic efficacy of the MRI examination. However, the biological effects of the higher field strength on spermatogenesis have not been investigated [22].

### 2.1. Normal Conventional MRI Findings

Normal testes are depicted as well-defined, ovoid, homogeneous structures, with signal intensity similar to that of the surrounding muscle, on T1WI. They appear hyperintense on T2WI, with thin, hypointense septa coursing radially and towards a low-signal-intensity band in the posterior aspect of the testis—the mediastinum testis. The T2 hyperintensity of the normal testicular parenchyma is ideal for the detection of intratesticular masses. The tunica albuginea is seen as a thin hypointense rim around the testis, best detected via T2WI (Figure 1) [22,26,29]. The normal epididymis is slightly inhomogeneous, being isointense to the testis on T1WI, and mostly hypointense on T2WI (Figure 1c). The spermatic cord is better depicted on coronal T2WI, and appears hyperintense with internal flow voids, due to the presence of fat and vessels. The vas deferens demonstrates a dark wall and a bright lumen on T2WI. A small hydrocele is often seen as a hyperintense rim surrounding the testis on T2WI, and represents a normal finding [26,29,35].

### 2.2. Diffusion-Weighted Imaging

DWI is a functional imaging technique based on the measurement of random diffusion movements of water molecules. Tissue cellularity and the presence of intact cell membranes largely determine the impedance of diffusion. This is quantitatively assessed by calculating the ADC, using different *b*-values. ADC is displayed as a parametric map, and expressed in units of mm^2^/s. DWI has a short acquisition time, and does not require contrast agents or expensive and complex hardware [50,51,52,53,54,55,56,57].

The testis’ structural complexity—including the presence of densely packed seminiferous tubules, which are lined with a compact fibroelastic connective tissue and interstitial stroma, containing fibroblasts, blood vessels, lymphatics, and Leydig cells—results in restricted diffusion. Normal testes appear hyperintense and slightly hypointense on high-*b*-value DWI and ADC maps, respectively, with an ADC ranging between 1.08 and 1.31 × 10^−3^ mm^2^/s [50,51,52].

DWI improves the diagnostic efficacy of MRI in the characterization of testicular masses. Testicular carcinomas often present with restricted diffusion and, therefore, are detected as hyperintense and hypointense on DWI and ADC maps, respectively, with a decrease in ADC (Figure 2), when compared to normal testes and benign testicular lesions. Tumors’ histological characteristics—such as increased cellularity, densely packed neoplastic cells, enlargement of the nuclei, and angulation of the nuclear contour—result in impedance of the motility of the water molecules [50,51,52,53,54,55,56,57].

The overall accuracy of conventional MRI, DWI alone, and DWI combined with conventional sequences in the characterization of testicular lesions has been reported to be 91%, 87%, and 100%, respectively [50]. An ADC cutoff of 0.99 × 10^−3^ mm^2^/s has been proven to be reliable for the characterization of testicular lesions, with 93.3% sensitivity and 90% specificity [52]. In a recent study, an ADC lower than 0.90 × 10^−3^ mm^2^/s was considered to be suggestive of the diagnosis of a testicular tumor [56]. In the same study, the authors found that adding DWI to the conventional MRI sequences increased the specificity and accuracy of the technique in the characterization of testicular masses by 9.8% and 3.2%, respectively [56].

Table 1 summarizes data from the literature on the diagnostic efficacy of DWI in the characterization of intratesticular masses.

### 2.3. Dynamic Contrast-Enhanced MRI

DCE-MRI is a functional imaging tool that provides information on tissue perfusion, vessel permeability, and the structure of the extracellular–extravascular space. Semi-quantitative analysis of DCE imaging data evaluates changes in signal intensity over time within a specific region of interest (ROI), and describes tissue enhancement by calculating parameters such as maximum enhancement, time to peak, wash-in rate, and washout. Early changes in the time–signal intensity (TSI) curve correlate with the blood flow, and late changes are mainly related to the extravascular accumulation of the contrast medium through capillary permeability [58].

DCE-MRI of the scrotum is strongly recommended when US results on testicular mass lesion characterization are equivocal [14]. Differences in TSI curves between normal testes, benign testicular lesions, and testicular tumors have been described [59,60,61,62,63,64]. Normal testes enhance moderately and homogeneously, with a gradual, progressive increase in enhancement throughout the dynamic scan (type I curve) (Figure 3) [59,60,61,62,63,64]. This behavior is probably related to an intact blood–testis barrier [65]. TGCTs typically present with a type III curve, which is early and avid enhancement, followed by a gradual washout of the contrast medium (Figure 4). Absence of contrast enhancement represents a specific sign for the characterization of the benign nature of testicular masses (type 0 curve). Alternatively, benign testicular lesions may enhance homogeneously or heterogeneously, with early, strong enhancement, followed by either a plateau or a further slight, progressive contrast enhancement (type II curve) (Figure 3) [62,63].

In a retrospective study of 27 testicular lesions, type I curves were detected in all (100%) normal testes, type III curves were seen in all (100%) TGCTs, and type II curves were found in 63.6% of benign lesions. The maximum time to peak, which is defined as the difference between the arrival of the contrast medium and the time of peak intensity, proved to be an independent predictor of malignancy in the same study [62].

Table 2 presents data from the literature on the role of DCE-MRI in the characterization of intratesticular masses.

### 2.4. New Advances

#### 2.4.1. Diffusion Tensor Imaging

DTI was developed based on DWI to depict the direction and speed of water molecules’ diffusion. Diffusion is essentially a three-dimensional process; therefore, molecular mobility in structured tissues may be anisotropic [65,66]. The normal testis is a structured organ, with the seminiferous tubules, septa, and vessels orientated towards the mediastinum testis [42]. DTI calculates both ADC and fractional anisotropy (FA), providing information not only on the amount of diffusion, but also on the anisotropy of diffusion, which may occur in an anisotropic manner (Figure 1). Tissue structural integrity and microstructural changes can be assessed by DTI [66,67].

Testicular lesions—both benign and malignant—often have increased anisotropy when compared to normal testes. Specifically, TGCTs usually present with higher FA compared to normal testicular parenchyma (Table 3; Figure 2 and Figure 4). Increases in the number of neoplastic cells, the amount of cell membranes, and intracellular viscosity, combined with a decrease in the extracellular space, may explain the anisotropy seen in testicular tumors [42].

#### 2.4.2. Magnetization Transfer Imaging

MTI is a technique that enables measurements beyond those in conventional MRI images, based on the interactions between free water protons, which are responsible for the conventional MRI signal and restricted protons, which are bound to proteins and macromolecules. Exchange and cross-relaxation of magnetization between these two proton pools creates the MT effect. The MTI phenomenon is quantified by the MT ratio (MTR) [68].

The macromolecular content of normal testes is responsible for an MTR of 46%. The main contributing factors to the MT effects of the normal testicular parenchyma include the collagenous content and the presence of ample smooth endoplasmic reticulum—a membranous structure that plays an important role in the synthesis of testosterone [43].

MTI represents a supplementary MRI tool for the characterization of testicular masses. TGCTs often have high MTR (Table 4, Figure 5) when compared to normal testes and benign testicular lesions—a finding that may be related to tumoral heterogeneity and the presence of fibrovascular septa in cases of seminomas, and to the presence of hemorrhagic foci in cases of nonseminomas [43].

#### 2.4.3. Proton Magnetic Resonance Spectroscopy

1H-MRS provides information about the biochemical milieu of normal tissues and pathological entities [69,70]. The prominent metabolic peaks of the 1H-MR spectrum of normal adult testes include choline, creatine, myo-inositol, and lipids (Figure 1f) [71]. Based on preliminary data, a decrease in choline levels is detected in testicular malignancies (Figure 2e) [44].

#### 2.4.4. Volumetric ADC Histogram Analysis

Volumetric ADC histogram analysis is a quantitative method that describes the statistical information contained in an image. Most histogram analyses use descriptive parameters—including mean ADC; standard deviation; mode, maximum, and minimum ADC; percentiles of ADC; kurtosis; skewness; and entropy—representing the first-order statistical properties of the image. This technique uses a volume of interest (VOI) covering the entire lesion, which assesses diffusion properties and heterogeneity [72,73]. Volumetric ADC histogram analysis performs better than mean ADC, which is based on an ROI placed on the most representative part of the lesion and, therefore, not reflecting histologic heterogeneity, and possibly resulting in measurement sampling errors and subjective bias [72,73]. Changes in histogram shape and asymmetry reflect microstructural and functional differences in tumor composition, and are correlated with the histological grade, TNM, tumor aggressiveness, and therapeutic response [72,73].

In a recently published retrospective study of 61 histologically proven testicular lesions, the minimum ADC and the 10th percentile of ADC were reduced in testicular tumors compared to benign testicular lesions. The minimum ADC had the highest diagnostic performance in characterizing testicular lesions, with 81.4% sensitivity and 77.78% specificity [45]. The lower percentiles of ADC better represent the highest cellular areas within tumors, explaining their diagnostic superiority in the characterization of malignancies (Table 5) [72,73].

#### 2.4.5. MRI-Based Radiomics

Radiomics is an emerging quantitative approach to imaging that aims to provide information beyond that which can be perceived from human imaging interpretation alone, by means of advanced mathematical analysis [74,75]. Texture analysis represents a technique used for the quantification of tissue heterogeneity. Histogram analysis features and intra-perinodular textural transition analysis features based on manual segmentation of testicular lesions on T2WI proved helpful in the differential diagnosis between benign and malignant testicular lesions (Table 6) [47].

## 3. Characterization of Testicular Masses: Benign versus Malignant

Although the majority of testicular masses are malignant, a possible characterization of the benign nature of various testicular entities—such as cysts, tubular ectasia of the rete testis, fibrosis, hematoma, segmental testicular infarction, intratesticular lipoma, Leydig cell hyperplasia, and adrenal rest tumors—based on imaging features improves patient management, decreases the number of unnecessary radical surgical explorations, and reduces healthcare costs [7,9,14,17,22,24,25,26,27,28,29,30,31,32,33,34,35,36,37,38]. In these cases, a conservative approach—including follow-up, biopsy, and testis-sparing surgery (TSS)—may be justified. Specifically, TSS is recommended in cases of small or indeterminate testicular tumors with negative tumor markers in the presence of a normal contralateral testis, in order to avoid the overtreatment of potentially benign lesions, and to preserve testicular function [8].

Based on the recommendations of the SPIWG, mpMRI of the scrotum represents a valuable problem-solving tool for the characterization of testicular masses in patients with equivocal US findings, or in cases in which US features are inconsistent with the clinical findings [14]. MRI provides useful information in terms of lesion location, as well as morphological and functional data, by demonstrating the presence of fat, fluid, blood products, fibrosis, myxoid and granulation tissue, diffusion restriction, and contrast-enhancing elements [7,9,14,17,22,24,25,26,27,28,29,30,31,32,33,34,35,36,37,38].

Conventional MRI—including T1WI, T2WI, and contrast-enhanced T1WI—had 100% sensitivity and 87.5% specificity in the differentiation between benign and malignant testicular masses, in a prospective study of 36 testicular lesions [76]. A multicenter retrospective review identified 34 patients (1.4%) examined with scrotal MRI following indeterminate clinical and US examination. MRI was accurate in the characterization of scrotal lesions in 91% of cases, and helped to improve the treatment provided by general urologists and urological oncologists in 56% and 50% of patients, respectively [36] (Table 7).

### 3.1. MRI Findings of TGCTs

TGCTs typical appear isointense on T1WI compared to normal testes. Areas of hemorrhage may be seen as intratumoral hyperintense foci on T1WI. These neoplasms are either detected with a low T2 signal, or are heterogeneous, with variable signal, on T2WI [7,19,24,76,77,78]. Testicular tumors often cause restricted diffusion and enhance heterogeneously after the administration of gadolinium, with a type III curve (Figure 2, Figure 4, Figure 5 and Figure 6) [52,53,62]. Areas of intratumoral necrosis are hyperintense on T2WI, with a lack of enhancement (Figure 6d). Tumor extension to the testicular tunicae, paratesticular space (Figure 7), and/or the spermatic cord confirms the diagnosis of malignancy [7,19,24,76,77,78].

### 3.2. Epidermoid Cysts

Testicular epidermoid cysts (ECs) represent a subtype of teratoma of the prepubertal type, included in the group of TGCTs unrelated to GCNIS [39]. The ability of preoperative imaging to suggest the diagnosis of an EC may prompt a conservative treatment, including TSS, instead of radical orchiectomy [8,79,80,81,82,83].

Histologically, EC is a squamous epithelium-lined unilocular cystic lesion, composed of concentric layers of laminated keratinous material, which are responsible for its characteristic imaging appearance [79,80,81,82,83]. MRI may provide supportive findings for the diagnosis of EC, and is recommended as a complimentary imaging examination in cases of equivocal US findings [14].

Typically, an EC is an oval encapsulated lesion surrounded by a hypointense halo on both T1WI and T2WI, which does not enhance after the administration of the contrast material. Alternating zones of high and low signal intensity on T2WI (“onion skin sign”), or a hyperintense central area (“target appearance”)—the latter corresponding to the paradox T1 hyperintensity of calcium depositions—are imaging findings suggestive of the diagnosis [79,80,81,82,83].

### 3.3. Testicular Lymphoma

Although uncommon, primary testicular lymphoma is the most common bilateral testicular neoplasm. It is an aggressive malignancy, and may infiltrate the epididymis, spermatic cord, and/or scrotal skin. Most testicular lymphomas are diffuse large B-cell lymphomas [18,28,41,56,84,85].

On MRI, these tumors are often hypointense on T2WI, enhance strongly and heterogeneously, and cause significant diffusion restriction (Figure 8) [56,85]. Ipsilateral lymphomatous epididymal involvement and contralateral testis infiltration may be better depicted on MRI.

## 4. Characterization of Testicular Tumors: Germ-Cell versus Sex-Cord Stromal Testicular Tumors

In recent years, the widespread use of scrotal US has resulted in a high incidence of asymptomatic, impalpable, small, and solid testicular tumors, detected as incidental findings. The prevalence of benign histology is approximately 80% in these cases, and LCT represents the most common pathology [86]. TSS for small US-detected, non-palpable intratesticular lesions is highly recommended to obtain a histological diagnosis. When a non-germ-cell tumor is suggested by frozen section examination (FSE), orchiectomy may be avoided [8,87]. Active surveillance through clinical and radiological follow-up, once the diagnosis is strongly suggested based on imaging findings, represents another alternative option for small LCTs [87].

Although no reliable imaging features yet exist, mpMRI may help in the characterization of LCTs, and especially in their differentiation from testicular seminomas [14,17,24,63,88,89,90,91]. An accurate MRI characterization of LCTs has been reported in up to 93% of cases.

LCTs are often small, sharply defined, isointense on T1WI, and markedly hypointense on T2WI, with strong, early, homogeneous enhancement, followed by slow washout (Figure 9) [63,88,89,90,91]. Additional T2WI characteristics include the presence of capsular high signal intensity and a hyperintense central scar [88]. Ill-defined margins, mild T2 lesion hypointensity, mild T1 hyperintensity, and weak and gradual enhancement, without de-enhancement, is more often seen in seminomas [63,88,89,90,91]. Differences in semi-quantitative DCE-MRI parameters were found between LCTs and seminomas, with an increase in peak enhancement and wash-in rate, along with a decrease in time to peak, observed in LCTs [90].

Based on TSI curves, quantitative DCE data can also be calculated, including the volume transfer constant, rate constant, extravascular extracellular space volume fraction, and initial area under the curve in the first 60 s, providing valuable information on tumor angiogenesis [58]. Increases in the volume transfer constant, rate constant, and initial area under the curve have been reported in LCTs, confirming a typical pattern of enhancement for these neoplasms, which is an avid and rapid wash-in [90].

DCE-MRI parameters proved useful in the differentiation between benign testicular stromal tumors, malignant neoplasms, and BOTs in a study of 31 impalpable, incidentally discovered testicular tumors on sonography. Specifically, benign stromal tumors had a higher maximal relative enhancement, shorter time to peak, higher initial slope, and higher transfer constants when compared to malignancies and BOTs [63]. Multiparametric MRI findings in a series of 10 BOTs, discovered in asymptomatic infertile men, were as follows: well-defined intratesticular nodule, with low T2 signal, high ADC, and lack of enhancement [92].

In a recently published retrospective study, T2WI-based radiomics proved useful in the preoperative differentiation between TGCTs and sex-cord stromal tumors, with 86% diagnostic accuracy (Table 8) [49].

### Sertoli-Cell Tumors

SCTs are less common than LCTs, accounting for about 1% of all testicular neoplasms. These neoplasms are rarely malignant [40]. TSS may be recommended, with subsequent radical orchiectomy if histology proves malignant [8].

A few MRI studies have described SCTs as homogenous lesions, isointense to normal testes on T1WI, with a variable T2 signal, hyperintense or hypointense, with strong, early, homogeneous enhancement, followed by rapid washout of the contrast medium [7,24,28,29,34,59,62,63,93].

## 5. Histological Characterization of TGCTs

Radical inguinal orchiectomy with removal of the entire testis, containing the tumor along with the spermatic cord, to the level of the internal inguinal ring, is the preferred treatment for all testicular tumors, and should be performed within a week of initial diagnosis, unless clinical indications require immediate chemotherapy [8].

In patients presenting with disseminated disease and/or life-threatening metastases, orchiectomy may be delayed. The first-line treatment for metastatic TGCTs mainly depends on the histology of the primary tumor [8]. MRI is recommended to differentiate between seminomas and NSGCTs in these cases [14]. MRI features have been found to closely correlate with the histopathological characteristics of TGCTs, with an accuracy of up to 91% [7,17,19,20,22,24,46,48,64,77,78].

Testicular seminoma is typically seen as a multinodular tumor, mainly homogeneous, with a low T2 signal. Intratumoral hypointense bands on T2WI are often detected, which enhance more than the remaining tumor, and correspond to fibrovascular septa in terms of pathology (Figure 2 and Figure 5) [77,78]. Conversely, nonseminomas often appear markedly heterogeneous on both T1WI and T2WI, with inhomogeneous contrast enhancement (Figure 4 and Figure 6)—findings that are mainly associated with the presence of hemorrhage and/or necrosis histologically. A hypointense T2 rim surrounding the tumor is more commonly seen on nonseminomas, corresponding to a fibrous capsule in terms of histology (Figure 6) [77,78]. Moreover, seminomas usually cause significant diffusion restriction compared to nonseminomas (Figure 2, Figure 4, Figure 5 and Figure 6)—a finding that is related to the presence of densely packed uniform cells, with abundant cytoplasm and large nuclei, surrounded by fibrous septa containing eosinophils and lymphocytes in seminomatous tumors. An ADC cutoff of 0.68 × 10^−3^ mm^2^/s is reliable for the histological characterization of TGCTs [56,64].

Whole-tumor ADC histogram analysis may also help in the characterization of TGCTs. Decreases in the median, mean, minimum, maximum, and percentiles of ADC, and increases in kurtosis and skewness, have been observed in testicular seminomas compared to nonseminomas, reflecting the compact, uniform histological composition of seminomatous tumors, in contrast to the heterogeneous, cystic appearance of NSGCTs [46]. Among volumetric ADC histogram parameters, the 10th percentile of ADC yielded the highest diagnostic performance, with 100% sensitivity and 92.86% specificity [46]. Recently, a T2-based radiomics signature had 90% sensitivity and 100% specificity in the histological differentiation of TGCTs (Table 9) [48].

## 6. Local Staging of TGCTs

In patients with TGCTs, radical orchiectomy is the treatment of choice, as pathological studies report multifocal and/or adjacent GCNIS in 20–30% of cases [8,94,95]. TSS, when feasible, can be attempted in special circumstances, including synchronous bilateral testicular tumors and tumors in a solitary testis [8]. TSS, in properly selected cases, has the advantage of better preserving endocrine function and fertility, with a reduced impact on sexual and psychological aspects, and satisfactory oncological outcomes [8,96]. TSS should be followed by FSE, which has been proven to be highly accurate and consistent with the final histological diagnosis [97,98]. However, in cases of discordance between FSE and final histopathology, delayed orchiectomy is justified [8].

Accurate preoperative knowledge of the local extent of TGCTs is important in patients who are candidates for TSS. MRI is recommended for local staging of TGCTs in these cases [14]. The technique performs well in the preoperative evaluation of the local staging of TGCTs. Accurate information regarding tumor dimensions and possible invasion of the rete testis, testicular tunicae, paratesticular space, and/or spermatic cord is assessed by MRI. Tumor pseudocapsules detected as hypointense halos surrounding lesions on T2WI have been described as a feature facilitating enucleation [14,17,22,24,25,28,32,35,76].

Regarding the local staging of the primary tumor (pT) in testicular malignancies, the 2016 TNM classification of the International Union Against Cancer includes the following stages: pT1—tumor limited to the testis and epididymis, without vascular/lymphatic invasion; tumor may invade the tunica albuginea, but not the tunica vaginalis; pT2—tumor limited to the testis and epididymis, with vascular/lymphatic invasion or tumor extending through tunica albuginea, with involvement of the tunica vaginalis; pT3—tumor invades the spermatic cord, with or without vascular/lymphatic invasion; pT4—tumor invades the scrotum, with or without vascular/lymphatic invasion [8,99]. MRI criteria for the evaluation of the local extent of testicular tumors are as follows: pT1—intratesticular tumor surrounded by a rim of normal testicular parenchyma and/or intact testicular tunicae, which are detected as a continuous hypointense rim in the periphery of the testis; pT2—intratesticular tumor invading the testicular tunicae, detected as a disruption of the T2 hypointense peripheral halo, with or without the presence of a paratesticular mass; pT3—enlargement, heterogeneity, and enhancement of the spermatic cord, due to neoplastic invasion; pT4—neoplastic invasion of the scrotal wall [76]. The rate of correspondence between MRI and histological diagnosis in local staging of TGCTs has been reported to be 92.8% (Table 10) [76].

## 7. Conclusions

Multiparametric MRI of the scrotum has emerged as a valuable, second-line diagnostic tool for imaging of testicular tumors. This technique is recommended for the characterization of testicular masses when conventional US findings are indeterminate. MRI is accurate in the differentiation between benign and malignant testicular masses, helping to avoid unnecessary radical surgical procedures. Scrotal MRI estimates the local extent of testicular carcinomas, allowing for surgical planning in candidates for testis-sparing surgery. MRI features are closely correlated with the histopathological characteristics of testicular germ-cell tumors. Therefore, this technique is recommended in the rare cases presenting with disseminated disease and/or life-threatening metastases, where chemotherapy represents the primary treatment, to differentiate between seminomas and nonseminomatous germ-cell tumors. Although still under investigation, multiparametric MRI of the scrotum may provide valuable information in differentiating between germ-cell and sex-cord stromal tumors—especially in cases of small, impalpable, incidentally detected testicular tumors.

Functional MRI techniques—including diffusion tensor imaging, magnetization transfer imaging, proton MR spectroscopy, volumetric ADC histogram analysis, and MRI-based radiomics—may provide new insights in the understanding of testicular tumors in the future.

## Figures and Tables

**Figure 1 cancers-14-03912-f001:**
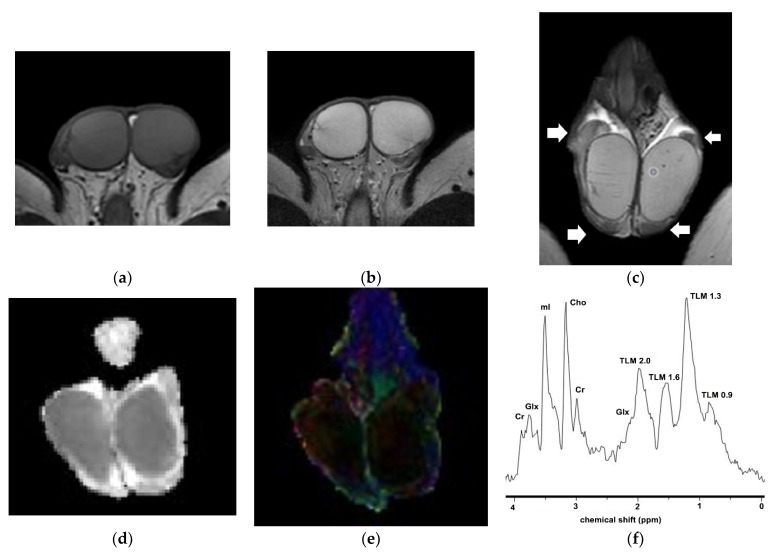
Normal MRI findings in a 23-year-old man: Axial (**a**) T1WI and (**b**) T2WI show normal testes as homogeneous ovoid structures, with intermediate T1 and high T2 signals. The tunica albuginea appears as a thin hypointense rim surrounding the testes, best seen on T2WI. (**c**) Coronal T2WI shows bilateral epididymal heads and tails (arrows), which are mostly hypointense compared to the normal testicular parenchyma. Thin, hypointense septa are detected within the right testis. A small bilateral hydrocele is also seen (normal finding). Coronal (**d**) ADC and (**e**) color-coded FA maps show bilateral normal testes. The mean ADC and FA of the normal right testis are 1.07 × 10^−3^ mm^2^/s and 0.06, respectively. (**f**) 1H-MR spectrum of the right testis. Prominent metabolic peaks include the following: Cr, creatine; Glx complex, glutamine and glutamate; mI, myo-inositol; Cho, choline; TLM, total lipids and macromolecules resonating at 2.0 ppm, 1.6 ppm, 1.3 ppm, and 0.9 ppm (parts per million).

**Figure 2 cancers-14-03912-f002:**
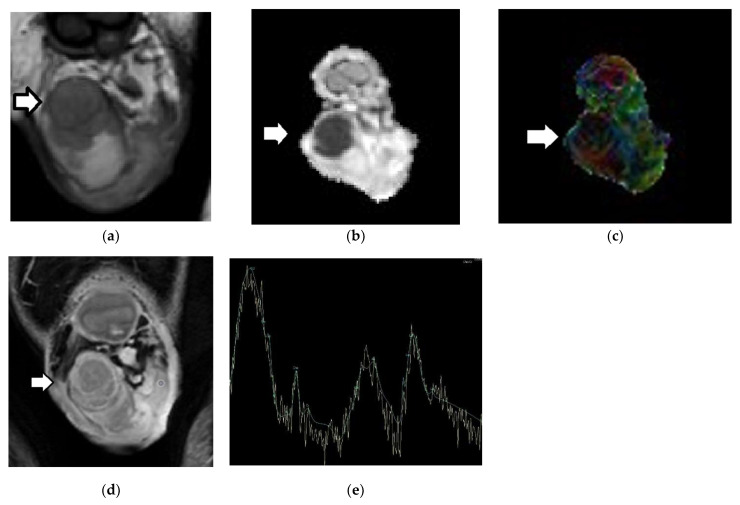
Right testicular seminoma in a 48-year-old man: The patient had a history of left radical orchiectomy due to seminoma. The incidence of metachronous TGCT in the contralateral testis is approximately 1–5% in men affected by TGCTs. Coronal (**a**) T2WI, (**b**) ADC, (**c**) color-coded FA map, and (**d**) subtracted DCE images demonstrate a multinodular right intratesticular tumor (arrow)—mainly homogeneous—of low T2 signal. Intratumoral septa are seen as hypointense bands on T2WI, enhancing more than the remaining tumor. The carcinoma appears with low signal on the ADC map, due to restricted diffusion. The ADC and FA of testicular seminoma are 0.51 × 10^−3^ mm^2^/s and 0.15, respectively. (**e**) 1H-MR spectrum of testicular seminoma shows a significant decrease in choline peaks (Cr, creatine; Glx complex, glutamine and glutamate; mI, myo-inositol; Cho, choline; Lac, lactate; Lip: lipids).

**Figure 3 cancers-14-03912-f003:**
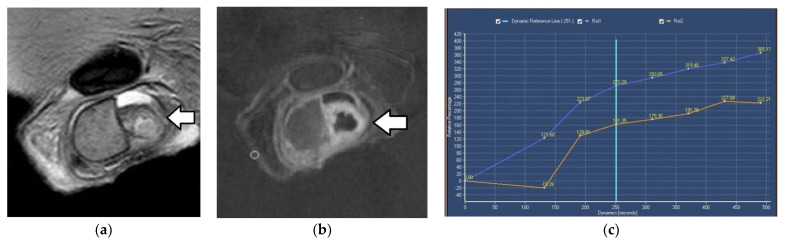
Right acute epididymo-orchitis in a 53-year-old man: Coronal (**a**) T2WI and (**b**) subtracted DCE images depict enlargement, T2 heterogeneity, and inhomogeneous enhancement of the right epididymis (arrow) and the ipsilateral testis. (**c**) TSI curves of the right acute orchitis (type II, purple) and the normal contralateral testis (type I, orange).

**Figure 4 cancers-14-03912-f004:**
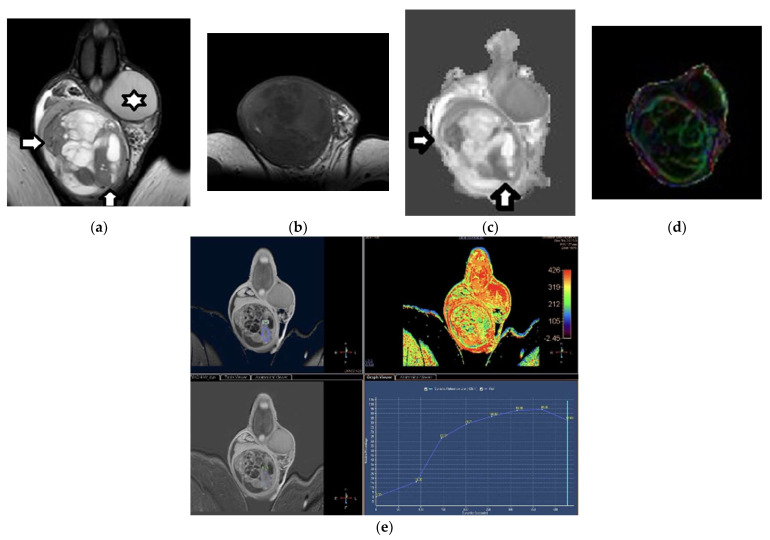
Right mixed TGCT (seminoma, embryonal carcinoma, and teratoma) in a 23-year-old man: (**a**) Coronal T2WI shows a large, extremely heterogeneous testicular tumor, replacing the right testis. The neoplasm has hyperintense and hypointense parts (arrows), the latter corresponding to the seminomatous histological component. (**b**) Transverse T1WI image depicts the tumor’s inhomogeneity. Coronal (**c**) ADC and (**d**) color-coded FA maps. Note: the seminomatous part (arrows) causes significant restricted diffusion (mean ADC: 0.78 × 10^−3^ mm^2^/s). The neoplasm shows increased anisotropy. The mean FA of the seminomatous and nonseminomatous parts of the tumor is 0.16 and 0.12, respectively—higher than that of the contralateral normal testis (star, FA: 0.04). (**e**) Coronal DCE imaging and TSI curve show the neoplasm enhancing strongly and heterogeneously, with a type III curve.

**Figure 5 cancers-14-03912-f005:**
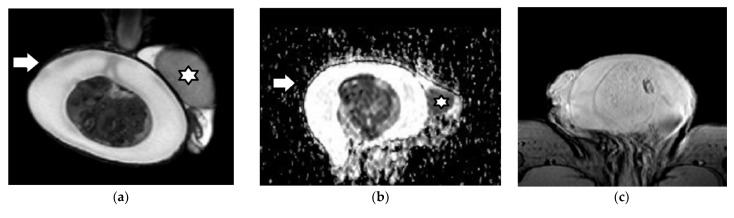
Right testicular seminoma in a 30-year-old man: (**a**) Coronal T2WI demonstrates a large, mainly hypointense testicular tumor, almost replacing the right testis. (**b**) Axial ADC map shows the tumor causing restricted diffusion. The mean ADC of testicular seminoma is 0.69 × 10^−3^ mm^2^/s. Axial 3-dimensional gradient-echo images acquired without (**c**) and with (**d**) the application of the magnetization transfer pulse. The testicular tumor appears hypointense. The MTR of the testicular seminoma is 57.8 %, compared to the MTR of 45.5 % for the contralateral normal testis (star). (**e**) Coronal subtracted DCE image depicts the neoplasm enhancing inhomogeneously. A large ipsilateral hydrocele is also seen (arrow).

**Figure 6 cancers-14-03912-f006:**
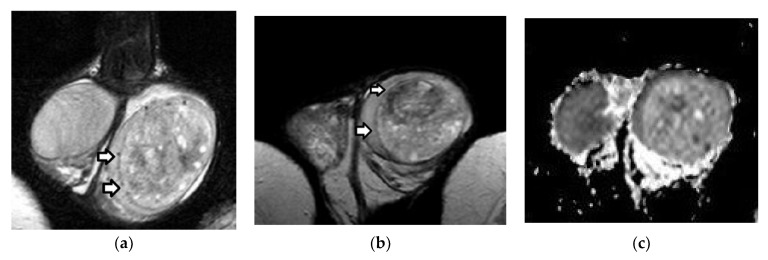
Left mixed TGCT (embryonal carcinoma, teratoma, and yolk sac tumor) in a 22-year-old man: T2WI in (**a**) coronal and (**b**) axial planes shows a large, heterogeneous left intratesticular tumor. The mass is surrounded by a hypointense halo (arrows), proven to correspond to the presence of pseudocapsule in terms of pathology. (**c**) Transverse ADC map. The mean ADC of testicular carcinoma is 1.25 × 10^−3^ mm^2^/s. (**d**) Coronal subtracted DCE image depicts inhomogeneous tumor enhancement. A large non-enhancing intratumoral area (star) is seen, due to the histological presence of necrosis.

**Figure 7 cancers-14-03912-f007:**
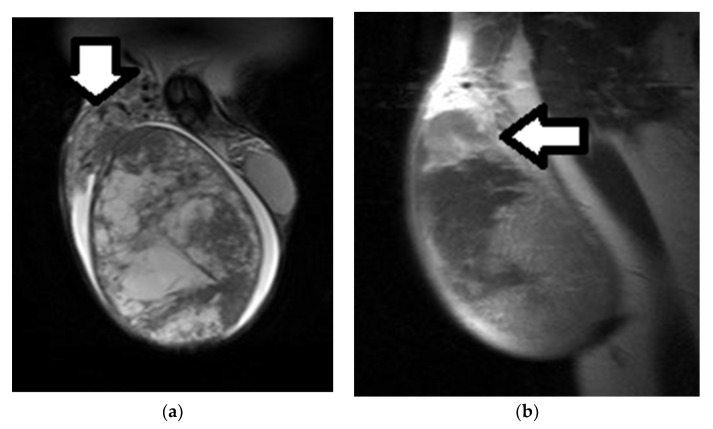
Right testicular seminoma in a 27-year-old man: (**a**) Coronal T2WI and (**b**) sagittal contrast-enhanced T1WI images depict a large, heterogeneous tumor, replacing the right testis and extending into the paratesticular space (arrow). Large seminomas may have areas of necrosis, which appear hyperintense on T2WI, not enhancing after gadolinium administration.

**Figure 8 cancers-14-03912-f008:**
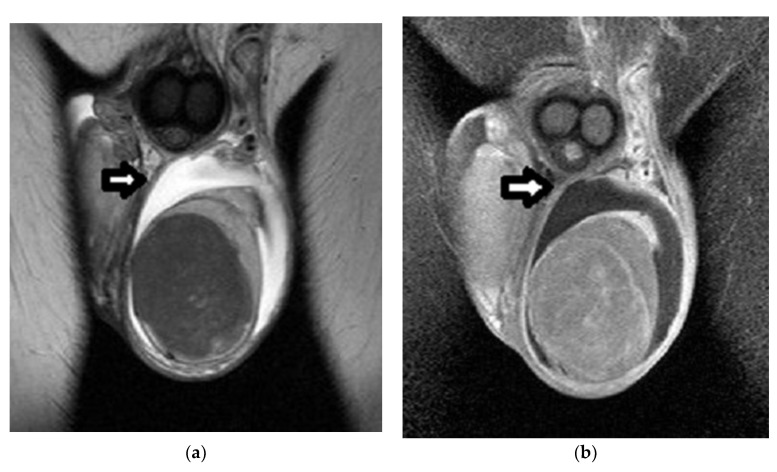
Right primary diffuse large B-cell testicular lymphoma in a 52-year-old man: Coronal (**a**) T2WI and (**b**) post-contrast T1WI images depict a multinodular right testicular mass. The tumor is mainly homogeneous, and has a low T2 signal, enhancing inhomogeneously after the administration of gadolinium. A moderate hydrocele (arrow) is also seen ipsilaterally.

**Figure 9 cancers-14-03912-f009:**
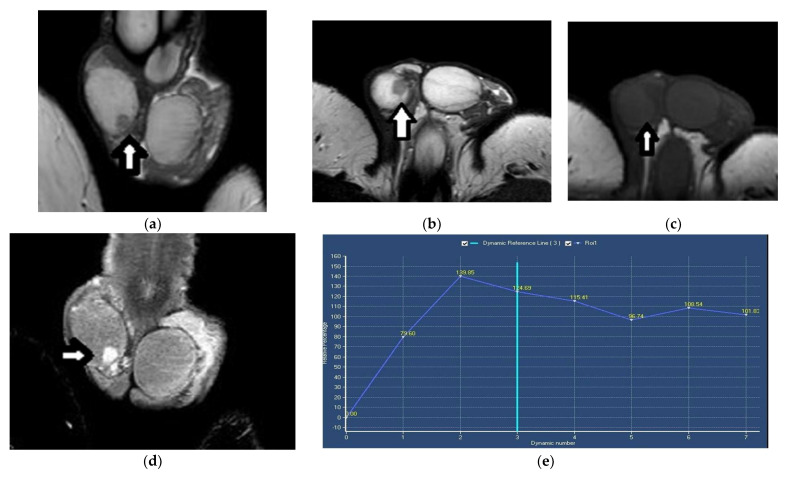
Right testicular Leydig-cell tumor in a 30-year-old man: (**a**) Coronal and (**b**) axial T2WI show a small, rounded, well-circumscribed, mainly hypointense intratesticular lesion (arrow) compared to the bright normal testicular parenchyma signal. (**c**) On axial T1WI images, the tumor (arrow) is indistinguishable from the normal testis. (**d**) Coronal DCE imaging and (**e**) TSI curve show a rapidly and strongly enhancing neoplasm with early washout and a type III curve.

**Table 1 cancers-14-03912-t001:** Diagnostic performance of diffusion-weighted imaging (DWI) in the characterization of testicular masses (PPV: positive predictive value; NPV: negative predictive value; ADC: apparent diffusion coefficient).

First Author (Year)	Study Aims	MRI Scanner	*b* Values (s/mm^2^)	Main Findings	Study Limitations
Tsili et al. (2012) [50]	To evaluate the diagnostic performance of DWI in the characterization of scrotal lesions	1.5 T	0, 900	Diagnostic performance of DWI alone: sensitivity, 85.7%; specificity 88.8%; PPV, 92.3%; NPV, 80%; accuracy, 87% Diagnostic accuracy of DWI + conventional imaging: 100%	Retrospective nature Small number of masses Consensus reading of MRI findings
Algebally et al. (2015) [52]	To assess the diagnostic value of adding DWI to conventional MRI in the characterization of scrotal lesions	1.5 T	400, 800	Diagnostic accuracy of DWI + conventional imaging: 100% ADC cutoff ≤ 0.99 × 10^−3^ mm^2^/s in the characterization of intratesticular masses: sensitivity, 93.3%; specificity, 90%; PPV, 87.5%; NPV, 94.7%	Not all histologic types of intratesticular masses included
Somnez et al. (2012) [53]	To determine the effectiveness of DWI in the characterization of testicular masses	1.5 T	0, 500, 1000	Diagnostic performance of lesion signal intensity on DWI: sensitivity, 90%; specificity, 60%; PPV, 81%, NPV, 75%; accuracy, 73%	Small number of masses
Wang et al. (2021) [56]	To assess the feasibility of conventional MRI features combined with ADC values for the differential diagnosis of testicular tumors	3.0 T	50, 1000	Diagnostic performance of ADC: sensitivity, 70.6%; specificity, 71.7%; accuracy, 71.4% Diagnostic performance of ADC + conventional imaging: sensitivity, 94.1%; specificity, 89.1%; accuracy, 90.5% ADC < 0.90 × 10^−3^ mm^2^/s: diagnosis of malignancy	Retrospective nature Small number of masses Subjective assessment of conventional MRI findings
Liu et al. (2022) [57]	To characterize testicular lesions with low T2 signals using DWI (mumps orchitis versus seminoma)	3.0 T	0, 1000, 2000	ADC cutoff for differentiating seminoma from mumps orchitis: 0.54 × 10^−3^ mm^2^/s Diagnosis of seminoma: sensitivity, 99%; specificity, 31%	Retrospective nature Small number of masses Limited histological types Low specificity of the cutoff value

**Table 2 cancers-14-03912-t002:** Diagnostic performance of dynamic contrast-enhanced MRI (DCE-MRI) in the characterization of testicular masses (TSI: time–signal intensity).

First Author (Year)	Study Aims	MRI Scanner	Main Findings	Study Limitations
Reinges et al. (1995) [59]	To evaluate the possibility of a dynamic MRI technique for differentiating between benign and malignant testicular lesions	1.5 T	Higher maximum increase in signal intensity after contrast injection in malignant tumors compared to normal testes and benign lesions	
Watanabe et al. (2000) [61]	To evaluate testicular enhancement patterns in various scrotal lesions on subtracted DCE-MRI	1.5 T	TSI curve parameters (i.e., relative percentages of peak height and mean slope) helped in dividing testicular diseases into two groups: one group with no or decreased enhancement, and the other group with increased contrast enhancement, including malignant testicular tumors and acute mumps orchitis	
Tsili et al. (2012) [62]	To assess the value of DCE subtracted MRI in distinguishing between benign and malignant testicular lesions	1.5 T	Strong association between the type of TSI curve and diagnosis: type I curve, 100% of normal testes; type II curve, 63.6% of benign intratesticular lesions; type III curve, 100% of malignant tumors; Relative maximum time to peak: independent predictor of malignancy	Retrospective nature Small number of masses Consensus reading of MRI findings

**Table 3 cancers-14-03912-t003:** Diagnostic performance of diffusion tensor imaging (DTI) in the characterization of testicular masses (ADC: apparent diffusion coefficient; FA: fractional anisotropy).

First Author (Year)	Study Aims	MRI Scanner	*b* Values (s/mm^2^)	Main Findings	Study Limitations
Tsili et al. (2012) [42]	To assess the efficacy of DTI in characterizing testicular pathology	1.5 T	0, 700	Lower ADC in malignancies compared to normal testes and benign testicular lesions; Higher FA in both malignancies and benign testicular lesions compared to normal testes	Small number of cases Consensus reading of MRI findings

**Table 4 cancers-14-03912-t004:** Diagnostic performance of magnetization transfer imaging (MTI) in the characterization of testicular masses (MTR: magnetization transfer ratio).

First Author (Year)	Study Aims	MRI Scanner	Main Findings	Study Limitations
Tsili et al. (2016) [43]	To assess the feasibility of MTR in characterizing various testicular lesions	1.5 T	Higher MTR in testicular carcinomas compared to benign lesions and normal testes	Retrospective nature Small number of masses Consensus reading of MRI findings

**Table 5 cancers-14-03912-t005:** Diagnostic performance of volumetric apparent diffusion coefficient (ADC) histogram analysis in the characterization of testicular masses.

First Author (Year)	Study Aims	MRI Scanner	*b* Values (s/mm^2^)	Main Findings	Study Limitations
Fan et al. (2020) [45]	To evaluate the role of volumetric ADC histogram analysis in discriminating between benign and malignant testicular masses	3.0 T	50, 100, 500, 1000	Lower minimum ADC and 10th percentile ADC in testicular malignancies compared to benign lesions; Diagnostic performance of minimum ADC: sensitivity, 81.40%; specificity: 77.78%	Small number of masses

**Table 6 cancers-14-03912-t006:** Diagnostic performance of MRI-based radiomics in the characterization of testicular masses (Ipris: intra-perinodular textural transition; T2WI: T2-weighted imaging).

First Author (Year)	Study Aims	MRI Scanner	Main Findings	Study Limitations
Zhang et al. (2021) [47]	To compare the performance of histogram analysis and Ipris features in distinguishing between benign and malignant testicular lesions	3.0 T	Differences in three histogram and nine Ipris features between benign and malignant lesions; Diagnostic performance of energy, total energy, and Ipris_shell1_id_std: sensitivity, 74.6%, 50.9%, and 65.5%, respectively; specificity: 72.0%, 96.0%, and 76.0%, respectively	Small number of masses Only T2WI data used Literature on Ipris limited to one study

**Table 7 cancers-14-03912-t007:** Diagnostic performance of conventional MRI in the characterization of testicular masses (PPV: positive predictive value; NPV: negative predictive value; US: ultrasonography; ADC: apparent diffusion coefficient).

First Author (Year)	Study Aims	MRI Scanner	*b*-Values (s/mm^2^)	Main Findings	Study Limitations
Mohrs et al. (2012) [32]	To evaluate the diagnostic value of MRI in the care of patients with suspected scrotal disorders	1.5 T		Diagnostic performance of MRI in the classification of scrotal lesions: sensitivity, 95%; specificity: 92%; PPV, 97%; NPV, 91%; accuracy, 97%	Retrospective nature Heterogeneous data, not always histologically confirmed Consensus reading of MRI findings
Serra et al. (1998) [36]	To determine the diagnostic utility and net cost of MRI in the management of clinically and sonographically inconclusive scrotal lesions	1.5 T		Diagnostic accuracy of US versus MRI: 29% and 91%; Improvement of the management plan with the addition of MRI: 56% of cases for the general urologist and 50% of cases for the urologic oncologist; Overall net cost savings: USD 543 per patient for the urological oncologist; USD 730 per patient for the general urologist; USD 3833 per patient originally scheduled for surgery	Retrospective study, not randomized; unclear whether the results can be generalized to other clinics
Muglia et al. (2002) [37]	To investigate the utility of MRI after inconclusive sonography in the evaluation of scrotal disease	0.5 T and 1.5 T		MRI: provided additional and correct information (compared with US) in 82.1% of cases MRI: great concordance with the final diagnosis in cases of testicular malignancies	MRI not performed in all pathologies Absence of histological confirmation in all cases Various US machines Only conventional US used
Wang et al. (2021) [56]	To explore the feasibility of conventional MRI features combined with ADC values for the differential diagnosis of testicular tumors	3.0 T	50, 1000	Diagnostic performance of ADC alone: sensitivity, 70.6%; specificity, 71.7%; accuracy, 71.4% ADC cutoff < 0.90 × 10^−3^ mm^2^/s: diagnosis of malignancy Diagnostic performance of conventional MRI model alone: sensitivity, 94.1%; specificity, 79.3%; accuracy, 87.3% Diagnostic performance of ADC + conventional MRI model: sensitivity, 94.1%; specificity, 89.1%; accuracy, 90.5%	Retrospective nature Small number of masses Subjective assessment of conventional MRI features
Tsili et al. (2010) [76]	To assess the role of MRI in the preoperative characterization of testicular neoplasms	1.5 T		Diagnostic performance of MRI in the characterization of malignant testicular tumors: sensitivity, 100%; specificity 87.5%; PPV, 96.5%; NPV, 100%; accuracy, 96.4%	Retrospective nature Small number of masses Consensus reading of MRI findings No direct comparison between sonographic and MRI findings

**Table 8 cancers-14-03912-t008:** Diagnostic performance of MRI in the differential diagnosis between germ-cell and non-germ-cell testicular tumors (T2WI: T2-weighted imaging; TPR: true positive rate; DCE: dynamic contrast-enhanced; BOTs: burned-out tumors; LCTs: Leydig-cell tumors; CEUS: contrast-enhanced US; ROI: region of interest; DWI: diffusion-weighted imaging; ADC: apparent diffusion coefficient; PPV: positive predictive value; NPV: negative predictive value; T1PC: T1 post-contrast; mpMRI: multiparametric MRI).

First Author (Year)	Study Aims	MRI Scanner	*b* Values (s/mm^2^)	Main Findings	Study Limitations
Feliciani et al. (2021) [49]	To assess the ability of MRI-based radiomics to differentiate between testicular germ-cell and non-germ-cell tumors	1.5 T		Diagnostic performance of T2WI-based radiomics: accuracy, 89%; TPR in predicting testicular germ-cell tumor, 94%; TPR in predicting testicular non-germ-cell tumor, 75%	Retrospective nature Small number of cases Class imbalance Internal validation, no external dataset
El Sanharawi et al. (2016) [63]	To evaluate DCE-MRI using qualitative, semi-quantitative, and quantitative parameters for the characterization of histologically proven, non-palpable, incidentally found intratesticular tumors	1.5 T		Benign stromal tumors had higher maximal relative enhancement, shorter time to peak, higher initial enhancement slope, and higher transfer constants compared to malignancies and BOTs Diagnosis of malignancies and BOTs: – Maximal relative enhancement: sensitivity, 78.95%; specificity, 100% – Time to peak: sensitivity, 89.47%; specificity, 83.33% – Initial slope: sensitivity, 89.47%; specificity, 83.33% – Transfer constant: sensitivity, 89.47%; specificity, 100% – Rate constant: sensitivity, 89.47%; specificity, 83.33% Diagnosis of benign stromal tumors: – Maximal relative enhancement: sensitivity, 100%; specificity, 78.95% – Time to peak: sensitivity, 83.33%; specificity, 89.47% – Initial slope: sensitivity, 83.33%; specificity, 89.47% – Transfer constant: sensitivity, 100%; specificity, 89.47% – Tate constant: sensitivity, 83.33%; specificity, 89.47%	Retrospective nature Not all histological types of testicular malignancies included No direct comparison between sonographic and MRI findings
Pozza et al. (2019) [87]	To analyze the conventional MRI findings of LCTs over a 10-year period			67.6% of LCTs had suggestive MRI findings	Selection bias
Manganaro et al. (2015) [89]	To evaluate the role of contrast-enhanced MRI in the identification of LCTs	1.5 T		Diagnostic performance of MRI in characterizing LCTs: sensitivity, 89.47%; specificity, 96.65% Diagnostic performance of MRI in characterizing malignancies: sensitivity, 95.65%; specificity, 80.95%; accuracy, 93%	Lesions previously evaluated with CEUS Semi-quantitative analysis/ROI placement affected by small tumor size
Manganaro et al. (2018) [90]	To explore the role of DCE-MRI using semi-quantitative and quantitative parameters and DWI in differentiating benign from malignant small, non-palpable, solid testicular tumors	1.5 T	0, 500, 1000	Higher percentages of peak enhancement, wash-in-rate, volume transfer constant, rate constant, and initial area under the curve, and shorter time to peak, in benign lesions compared to malignancies Higher percentages of peak enhancement, wash-in-rate, volume transfer constant, rate constant, and initial area under the curve, and shorter time to peak, in LCTs compared to seminomas Best diagnostic cutoff for identification of seminomas: volume transfer constant, ≤0.135 min^−1^; rate constant, ≤0.45 min^−1^; initial area under the curve, ≤ 10.96; wash-in-rate, ≤1.11; percentage of peak enhancement, ≤96.72; time to peak, >99 s All tumors had similar ADC	Majority of lesions: LCTs and seminomas; no other histological types Palpable masses > 1.5 cm excluded Small tumor size may influence MRI measurements 50% of cases referred for infertility
Khanna et al. (2021) [91]	To assess the diagnostic performance of multiparametric MRI in differentiating benign testicular stromal tumors from malignant testicular neoplasms (non-stromal and stromal)	1.5 T	0, 400, 800	Diagnostic performance of T2WI: sensitivity, 83%; specificity, 83%; PPV, 69%; NPV, 100%; accuracy, 85% Diagnostic performance of T2WI + DWI: sensitivity, 92%; specificity, 100%; PPV, 92%; NPV, 100%; accuracy, 95% Diagnostic performance of T2WI + DWI + T1PC: sensitivity, 92%; specificity, 100%; PPV, 93%; NPV, 100%; accuracy, 95%	Single-center study Retrospective nature Small number of masses
Rocher et al. (2017) [92]	Analysis of mpMRI findings in asymptomatic infertile men with pathologically confirmed BOTs	1.5 T	0, 800	mpMRI findings of BOTs: well-delineated nodule, low T2 signal, high ADC, and lack of contrast enhancement	Small number of cases

**Table 9 cancers-14-03912-t009:** Diagnostic performance of MRI in the differential diagnosis between seminomas and nonseminomas (ADC: apparent diffusion coefficient; T2WI: T2-weighted imaging; TPR: true positive rate; DCE: dynamic contrast enhancement).

First Author (Year)	Study Aims	MRI Scanner	*b* Values (s/mm^2^)	Main Findings	Study Limitations
Liu et al. (2019) [20]	To explore the utility of preoperative MRI for the differential diagnosis of testicular seminomas and nonseminomatous germ-cell tumors	3.0 T		Diagnostic accuracy of MRI in preoperative diagnosis of seminomas: 95%	Small number of tumors
Min et al. (2018) [46]	To assess the value of parameters derived from whole-lesion histograms of the ADC for the characterization of testicular germ-cell tumors	3.0 T	50, 1000	Lower median of 10th, 25th, 50th, 75th, and 90th percentiles, and lower mean, minimum, and maximum ADC, in seminomas compared with nonseminomas; Higher median of kurtosis and skewness of ADC in seminomas compared with nonseminomas; Diagnostic performance of 10th percentile ADC: sensitivity, 100%; specificity, 92.86%	Retrospective nature Small number of tumors Consensus reading of MRI data No subgroup analysis for different histological subtypes of nonseminomas No comparison of the diagnostic performance of each histogram parameter Variations in MRI protocol
Zhang et al. (2021) [48]	To evaluate the performance of T2WI-based radiomics signatures for differentiating between seminomas and nonseminomas	3.0 T		Diagnostic performance of T2WI-based radiomics signatures: sensitivity, 90%; specificity, 100%	Small sample size No independent validation cohort; internal validation instead Variations in MRI parameters (slice numbers and thickness)
Feliciani et al. (2021) [49]	To assess the ability of MRI-based radiomics to differentiate between seminomas and non-seminomatous germ-cell tumors	1.5 T		Diagnostic performance of T2WI-based radiomics: accuracy, 86%; TPR in predicting seminomas, 87%; TPR in predicting non-seminomatous germ-cell tumors, 86%	Retrospective nature Small number of cases Class imbalance Internal validation; no external dataset
Wang et al. (2021) [56]	To explore the feasibility of conventional MRI features combined with ADC values for the differential diagnosis of testicular tumors	3.0 T	50, 1000	ADC + cystic change: independent factors for differentiating testicular nonseminomas from seminomas; ADC + cystic change + T2 homogeneity: independent factors for differentiating nonseminomas from testicular lymphomas; ADC + intratumoral septa: independent factors for differentiating seminomas from lymphomas	Retrospective nature Small number of masses Subjective assessment of conventional MRI features
Tsili et al. (2015) [64]	To investigate the role of ADC values and DCE patterns in differentiating seminomas from nonseminomatous germ-cell tumors	1.5 T	0, 900	Lower ADC in seminomas compared to nonseminomas; ADC cutoff point of 0.68 × 10^−3^ mm^2^/s: sensitivity, 63.6%; specificity, 100% No differences in DCE parameters between seminomas and nonseminomas	Retrospective nature Small number of tumors Consensus reading of MRI findings Inter-scanner and intra-scanner variability
Tsili et al. (2007) [77]	To evaluate the role of MRI in the preoperative characterization of the histological type of testicular tumors and, more specifically, to differentiate seminomatous from nonseminomatous testicular neoplasms	1.5 T		Diagnostic accuracy of MRI: 91%	Retrospective nature Small number of tumors No direct comparison between sonographic and MRI findings Dynamic contrast-enhanced imaging not performed
Johnson et al. (1990) [78]	To evaluate the role of MRI in the preoperative differentiation between seminomatous and nonseminomatous testicular neoplasms	1.5 T		Diagnostic accuracy of MRI: 92.8%	

**Table 10 cancers-14-03912-t010:** Diagnostic performance of MRI in local staging of testicular germ-cell tumors.

First Author (Year)	Study Aima	MRI Scanner	Main Findings	Study Limitations
Tsili et al. (2010) [76]	To evaluate the role of MRI in the preoperative local staging of testicular neoplasms	1.5 T	Diagnostic accuracy of MRI: 92.8%	Retrospective nature Small number of tumors Consensus reading of MRI findings No direct comparison between sonographic and MRI findings

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
