# Peer review of "An Overview of the Role of Multiparametric MRI in the Investigation of Testicular Tumors"

_cancers, 2022, doi:10.3390/cancers14163912_

Round 1

Reviewer 1 Report

Thank you for the opportunity to review the manuscript entitled, “MRI and testicular tumors (Review).” The authors present a comprehensive overview of current MRI and testicular tumors knowledge. This is a challenging topic. The author team summarized the most relevant papers on the subject. They touch on the mpMRI protocol of the scrotum, characterization of tumors (germ cell vs. sex cord-stromal testicular tumors), MRI findings of testicular germ cell tumors (TGCTs), histologic description of TGCTs, and staging of TGCTs. The clinical topic is important, as the incidence rate for testicular cancer, despite not being common, has been increasing for several decades, with the reasons being very clear. Having good diagnostic tests is essential. 

However, I have some comments to improve the manuscript.

1. My biggest concern is that a similar review already exists: https://pubmed.ncbi.nlm.nih.gov/30386879/. I recommend the authors provide additional information on the likely implications of guidelines endorsing scrotal MRI as a supplemental imaging technique.

2. Another concern is the lack of synthesized summary tables that provide an overview of the most up-to-date literature on the finding. I suggest creating tables in each section that highlight and compare results in the literature and the possible limitations of each study. 

3. Another aspect that would be very useful to the review is a whole new section outlining the implications of accurate imaging on treatment. Does it result in better outcomes? How does it affect orchiectomy rates?

Author Response

Dear reviewer,

We appreciate the time and effort that you have dedicated to providing your valuable feedback on our manuscript. We are grateful for your insightful comments on our paper. We have been able to incorporate changes to reflect your suggestions.

Thank you for the opportunity to review the manuscript entitled, “MRI and testicular tumors (Review).” The authors present a comprehensive overview of current MRI and testicular tumors knowledge. This is a challenging topic. The author team summarized the most relevant papers on the subject. They touch on the mpMRI protocol of the scrotum, characterization of tumors (germ cell vs. sex cord-stromal testicular tumors), MRI findings of testicular germ cell tumors (TGCTs), histologic description of TGCTs, and staging of TGCTs. The clinical topic is important, as the incidence rate for testicular cancer, despite not being common, has been increasing for several decades, with the reasons being very clear. Having good diagnostic tests is essential. 

However, I have some comments to improve the manuscript 

  1. My biggest concern is that a similar review already exists: https://pubmed.ncbi.nlm.nih.gov/30386879/. I recommend the authors provide additional information on the likely implications of guidelines endorsing scrotal MRI as a supplemental imaging technique.
  • Response: We agree. We have, accordingly, provide additional information on the likely implications of guidelines endorsing scrotal MRI as a supplemental imaging technique.

  1. Another concern is the lack of synthesized summary tables that provide an overview of the most up-to-date literature on the finding. I suggest creating tables in each section that highlight and compare results in the literature and the possible limitations of each study.
  • Response: We agree. We have, accordingly, revised the paper by adding tables.

  1. Another aspect that would be very useful to the review is a whole new section outlining the implications of accurate imaging on treatment. Does it result in better outcomes? How does it affect orchiectomy rates?
  • Response: We agree. We have, accordingly, provide information on the implications of accurate imaging on treatment. We have chosen to include this information on sections referring to the primary indications of scrotal MRI.

Reviewer 2 Report

 MRI and testicular tumors (Review)

The title is short and lacking and needs to be more explicit. Pictoral review. The word review seems off. 

Simple summary – please don’t use abbreviations without presentation. e.g US

Introduction

Consider rephasing, as a precise treatment plan can also be obtained using UL.“Scrotal lesions can be re liably detected, precisely localized and characterized with MRI, allowing for more precise  treatment planning to be determined and surgical exploration or radical orchiectomy”

"Recently" -consider rephrasing. The reference was published back in 2018. 

The aim seem not comparable with a review as suggested in the title of the paper. Maybe instead of review, the title should include the word overview? Just a suggestion.

Material and method

Any thoughts on MRI field strengths and effect on image quality?

Can it be difficult to visualize both testes in the same MRI image due to heights differences in the scrotal sac?

Presentation of testicular tumors

Well choice of images.

Conclusion

Why is it so difficult to implement scrotal MRI? Many departments do not see MRI of the scrotum as any value. So to state that MRI has a strongly recommendation in tumors seems ambiguous.

Author Response

Dear reviewer,

We appreciate the time and effort that you have dedicated to providing your valuable feedback on our manuscript. We are grateful for your insightful comments on our paper. We have been able to incorporate changes to reflect your suggestions.

  • Comment 1: The title is short and lacking and needs to be more explicit. Pictoral review. The word review seems off.
  • Response: We agree. We have, accordingly, revised the title (An overview of the role of multiparametric MRI in the investigation of testicular tumors).

  • Comment 2: Simple summary – please don’t use abbreviations without presentation. e.g US
  • Response: We agree. We have, accordingly, revised the ‘Summary’ section.

  • Comment 3: Introduction

Consider rephasing, as a precise treatment plan can also be obtained using UL.“Scrotal lesions can be re liably detected, precisely localized and characterized with MRI, allowing for more precise  treatment planning to be determined and surgical exploration or radical orchiectomy”

  • Response: We agree. We have, accordingly, revised the manuscript (‘Introduction’ section, Lines 66-69).

  • Comment 4: "Recently" -consider rephrasing. The reference was published back in 2018. The aim seem not comparable with a review as suggested in the title of the paper. Maybe instead of review, the title should include the word overview? Just a suggestion.
  • Response: We agree. The word ‘recently’ has been deleted. We have, accordingly, revised the title (An overview of the role of multiparametric MRI in the investigation of testicular tumors).

  • Comment 5: Material and method. Any thoughts on MRI field strengths and effect on image quality?
  • Response: We agree. We have, accordingly, revised the manuscript (Lines 134-137).

  • Comment 6: Can it be difficult to visualize both testes in the same MRI image due to heights differences in the scrotal sac?
  • Response: We agree. We have, accordingly, revised the manuscript (Line 114).

  • Comment 7: Presentation of testicular tumors. Well choice of images.
  • Response: -

  • Comment 8: Why is it so difficult to implement scrotal MRI? Many departments do not see MRI of the scrotum as any value. So to state that MRI has a strongly recommendation in tumors seems ambiguous.
  • Response: We agree. The word ‘strongly’ has been deleted.

Scrotal MRI remains a complimentary imaging technique in the investigation of testicular masses. Scrotal US is diagnostic in most cases. The aim of this review is to familiarize both radiologists and clinicians with the additional role of multiparametric MRI of the scrotum in the investigation of testicular tumors.

Reviewer 3 Report

This study was reported the utility of MRI for the diagnosis of testicular tumor. Generally, this paper is well written. The reviewer thinks that this paper has useful information for readers. However, the reviewer would like to suggest some critiques to make this paper as follows.

Major revision

1.      On line 22, “For testicular mass …. In selected cases” is unclear. The authors should separate this part in multiple sentences.

2.      On line 78, GCTs should be spelled out.

Author Response

Dear reviewer,

We appreciate the time and effort that you have dedicated to providing your valuable feedback on our manuscript. We are grateful for your insightful comments on our paper. We have been able to incorporate changes to reflect your suggestions.

This study was reported the utility of MRI for the diagnosis of testicular tumor. Generally, this paper is well written. The reviewer thinks that this paper has useful information for readers. However, the reviewer would like to suggest some critiques to make this paper as follows. 

Major revision

  1. On line 22, “For testicular mass …. In selected cases” is unclear. The authors should separate this part in multiple sentences.
  • Response: We agree. We have, accordingly, revised the manuscript.

  1. On line 78, GCTs should be spelled out.
  • Response: We agree. We have, accordingly, revised the manuscript.

Round 2

Reviewer 1 Report

I believe the manuscript has been sufficiently improved to warrant publication in Cancers. All my comments and concerns have been adequately addressed.  However, please update the newly added tables so that there are no errors (including adding uppercase and punctuation).

Author Response

Agree. We have revised the Tables accordingly.